# Key functions for the transferability of a French school-based health promotion intervention

**Inès Sartori[1], Leslie Fonquerne[2], Camille Riera-Navarro◉[1]\*, Daphné Desmoriaux◉[1], Cyrille Delpierre[2], Véronique Régnier Denois[1]**

1 Health, Systemic Process (P2S) Research Unit, University of Lyon, University Jean Monnet, Saint-Etienne, France, 2 UMR1295 CERPOP–Inserm, Université de Toulouse, Toulouse, France

\* camille.riera-navarro@univ-st-etienne.fr

## Abstract

The "*Alliance for Health*" intervention is a French evidence-based initiative targeting teachers and after-school care staff, aimed at supporting the development of health-promoting environments for primary school pupils. Its initial implementation in 101 schools and 97 municipalities (randomly assigned to receive different levels of training and support) demonstrated promising outcomes. Yet, a critical question remains for broader implementation: under what conditions can the intervention achieve optimal transferability in new contexts? To address this issue, the key functions (*i.e.,* intervention's transferable components) were identified and distinguished from their forms (*i.e.*, adaptable activities that can be tailored to specific contexts), as conceptualized within the FIC (Functions/Implementation/Context) model. The key functions were co-constructed through an iterative, collaborative process, based on qualitative data (semi-structured interviews conducted with intervention recipients and institutional actors involved in implementation) and on knowledge exchange between project leaders and researchers from various disciplines, following established methods. Seven key functions were identified, covering 4 main dimensions: institutional support (macro-level social influences on health determinants); empowerment of teachers and after-school care staff (inter-professional training and provision of support and resources, micro and meso levels), adaptation to institutional and recipients' needs and constraints (micro level); and operational alignment and coherent coordination (meso and micro levels). These findings provide actionable recommendations to guide future implementation strategies, supporting the intervention's replication or scale-up by clarifying which components must remain consistent and which can be adapted to maintain fidelity while enhancing transferability.

## Introduction

The *Alliance* research project is a Population Health Intervention Research initiative designed to evaluate the outcomes and implementation of an intervention that

---

**Data availability statement:** Thank you for providing updated Data Availability information as requested. Based on the access information you provided, we've drafted for your approval the following proposed Data Availability statement, which includes all required declarations and conforms to the PLOS ONE formatting conventions: "Although anonymized, the data (qualitative interview transcripts in French) include detailed accounts from a small professional community, creating a risk of indirect re-identification. In addition, participants did not consent to unrestricted public sharing of full transcripts, and such deposition would not comply with ethics committee approval. Relevant supporting materials are available at the institutional address institutpresage@univ-st-etienne.fr (under the reference "Alliance 2025 qualitative data") upon reasonable request." Please confirm whether this information is accurate, and we will update your Data Availability statement accordingly on your behalf.

**Funding:** This work was funded by the French National Cancer Institute (Institut National Du Cancer; INCa-16818); the French Institute for Public Health Research (Institut pour la Recherche en Santé Publique; IReSP-AAP-2021-273199) as part of the 2021 Research Call on Health-Promoting Services, Interventions, and Policies (supported by CNAM, DGS, Inserm, MILDECA, and Santé publique France); the French Interministerial Mission for Combating Drugs and Addictive Behaviors (Mission Interministérielle de Lutte Contre les Drogues et les Conduites Addictives; 18HS176CF1, 18HS176CF2); and the Auvergne-Rhône-Alpes Regional Health Agency (Agence régionale de santé Auvergne-Rhône-Alpes; 19HS091), with funding allocated to Véronique Régnier Denois and Franck Chauvin (University Jean Monnet Saint-Etienne, Health, Systemic, Process Research Unit 4129). The funders had no role in study design, data collection and analysis, decision to publish, or preparation of the manuscript. The SO-RISP research network, with support from the French National Cancer Institute, received INCa-Cancéropôle GSO support. This research was funded by IReSP as part of the call for structuring actions for research on uses and addictions to psychoactive substances 2021 (IRESP-AAPSPA2021-V1-06). The funders had no role in study design, data collection and analysis, decision to publish, or preparation of the manuscript.

**Competing interests:** The authors declare no competing interests.

promotes health-supportive environments for children attending primary schools [1]. Although pupils are the final beneficiaries – with expected long-term impacts on their health literacy and psychosocial skills – the intervention targets their broader environment and operationalizes the Health-Promoting Schools approach. Target groups include municipalities (*e.g.,* after-school staff, elected officials) and professionals across various levels of the French National Education system: primary school teams (directors, teachers), and district pedagogical advisors, representatives (health advisors, inspectors). Grounded in evidence from the literature and co-developed with professionals from the National Education system, the *Alliance* intervention aims to structure health education practices and foster collective dynamics. To this end, the intervention is structured around three main axes: (1) training in health promotion and project management for the involved professionals; (2) support in deploying health promotion initiatives in schools; and (3) continuous provision of methodological and pedagogical resources [1]. The *Alliance* research project involves a multi-disciplinary research team, institutional partners (such as the Ministry of Education), public health professionals, and policymakers. It is rooted in a collaborative and participatory approach to co-constructing health promotion practices.

The *Alliance* intervention was first implemented in 101 schools (around 10,000 pupils) and 97 municipalities across four French departments of the Auvergne-Rhône-Alpes region (Cantal, Isère, Loire, Rhône). Among the participating schools, 90% were public (vs. 88% nationally [2]), 3% were located in REP+ priority education zones (vs. 8% nationally [3]), and the mean Social Position Index (SPI) was $110 \pm 10$ (vs. 105 nationally [4]). This index is a composite indicator developed by the French Ministry of Education that reflects the socio-economic characteristics of students' families based on their parents' socio-professional categories (min: 45, max: 185, higher values indicate more advantaged populations). Most schools (71%) were located in rural areas (to ensure sufficient pupil numbers across rural and urban settings, as rural schools are typically smaller). Schools were randomly assigned to an intervention group (n = 48), which received the intervention (training, support and resources) over three academic years (2019–2020, 2020–2021, 2021–2022), or a control group (n = 53), which did not receive the intervention during this period (phase 1). Preliminary findings from phase 1 suggested favorable evolutions in professionals' health education practices (*e.g.,* increased perceived competence, more structured practices, and a broader health promotion approach integrating protective factors such as psychosocial skills and health literacy) as well as in several pupil-reported indicators (*e.g.,* relationships with adults and peers, perceived school climate, psychosocial skills). These preliminary findings (not yet published) led to the decision to extend the intervention. In 2022–2023, the intervention was extended to the 53 control schools and municipalities (phase 2), thus offering an opportunity to examine its implementation in new contexts and identify the conditions that may support its transferability.

Indeed, a central question raised by Population Health Intervention Research (PHIR) is whether an intervention proven effective in one context can be transferred to another – where both its nature and implementation might be influenced by local

conditions [5–7]. Transferability cannot simply be assumed: evidence shows that interventions effective in one setting may fail in another, for instance due to differences in population characteristics, available resources, organizational structures, political contexts, or stakeholder involvement [6,8]. One way to support transferability in future implementations is to develop a better understanding of the intervention by describing it in detail across the different contexts in which it has been implemented [9]. This descriptive work is essential to identify the core components and processes that must remain consistent across contexts to preserve fidelity and effectiveness, while identifying where flexibility and contextual adaptation are both possible and necessary [6,10]. This aspect of transferability is particularly interesting, as it can inform decisions about adaptations for replication in new contexts and thus provide actionable guidance for stakeholders seeking to replicate or scale up the intervention.

Several approaches can be used to document interventions in detail, including reporting guidelines such as TREND [11], CReDECI 2 [12], and TIDieR [13], or to analyze the transferability of interventions, such as the ASTAIRE tool [10]. These approaches improve the quality of intervention descriptions or help compare implementation contexts, but they do not primarily aim to distinguish which components of an intervention should remain stable across contexts and which may be adapted to support transferability across settings. The "Key Functions/Implementation/Context" (FIC) model was developed to address this issue by proposing to deconstruct interventions into: (i) *key functions* (the essential processes through which an intervention achieves its objectives), potentially transferable to other contexts, (ii) the different *forms* these functions may take across contexts, and (iii) the implementation context of the intervention [14–17]. This approach is based on the assumption that an intervention's transferability (including its potential to reproduce effects) relies, among other factors, on the transfer of its key functions (a necessary foundation for supporting implementation in new contexts), while allowing their forms (*e.g.,* specific activities or tools) to be adapted to local conditions (Because PHIR interventions are implemented in heterogeneous settings, they cannot be reproduced identically. Adaptations are therefore not considered deviations, but rather necessary conditions for maintaining the intervention's underlying processes across contexts [8,9].). Accordingly, it aims not to ensure intervention effectiveness (*i.e.,* achieving targeted outcomes), but to inform and support transfer (*i.e.,* successful implementation) in new contexts [14].

This research aims to identify the key functions of the *Alliance* intervention and to distinguish them from their forms, as conceptualized within the FIC model (without applying the model in its entirety), with the objective of informing and supporting its transferability, and providing evidence-based guidance for future implementation.

## Materials and methods

The *Alliance* intervention was described and analyzed via the key functions (as conceptualized within the FIC model [15,16]), based on qualitative data collected during phases 1 and 2 of the *Alliance* research project.

### Key functions as conceptualized within the FIC model: Method and terminology

The FIC model provides a framework to deconstruct interventions in order to determine which core functions can be transferred across contexts to support the intervention's logic across contexts – *key functions* – and which activities – *forms* – must be adapted [15–17]. This approach enables a deeper understanding of interventions for transfer to new contexts. As conceptualized within the FIC model, the present analysis focused on identifying the following components:

- *Key functions* are the essential mechanisms and steps through which an intervention achieves its objectives, and they may be consistent across different implementation contexts [16]. A key function represents a potential component of the overall intervention strategy. Distinguishing between key functions and forms challenges the traditional notion of intervention "fidelity," based on the assumption that an intervention can remain faithful to its key functions even when the specific activities or forms differ across contexts [17]. Key functions can influence health determinants at multiple levels, including micro-, meso-, and macro-social dimensions [18];

- *Forms* represent the translation of key functions into concrete activities [16]. *Theoretical forms* refer to activities designed by project leaders for implementation, whereas *observed forms* emerge from the actual adaptation and evolution of these activities in different (school and municipal) contexts. Unlike key functions, forms may differ depending on the context.

- *Contextual factors* are not passive "backdrops"; they actively shape the underlying processes that determine how and why an intervention is effective. Because contexts are complex and unique [19], it is essential to consider them alongside the forms of the intervention.

This study focused on identifying the key functions and forms of the *Alliance* intervention to inform and support its transferability for future implementations. It did not involve applying the FIC model in its entirety. As such, social inequalities in health were not explicitly considered in the present study, although they are an important concern within PHIR and constitute an important dimension of the FIC model [20].

## Participants

Qualitative data were collected through semi-structured interviews conducted with professionals from schools and municipalities involved in the *Alliance* intervention during phase 1 (2019–2022) or phase 2 (2022–2023). A wide range of stakeholders were interviewed, including intervention recipients (teachers and after-school care staff), and institutional actors involved in implementation, such as representatives from the French National Education system (health advisors at the rectorate and departmental levels, district pedagogical advisors), and municipal authorities (elected officials, employees responsible for school affairs).

All schools and municipalities that participated in phases 1 or 2 were contacted for recruitment (from 20 March to 15 April 2022 for phase 1, from 15 March to 15 June 2023 for phase 2). In schools, interviews were primarily conducted with school directors rather than with teachers, as directors were the main contacts for the project and held the school's generic contact email address. Regarding representatives of the French National Education system, all health advisors who had been designated as *Alliance* project referents (one per department and one per academy) were invited to participate. The total number of interviews was predefined based on the number of schools and municipalities that had implemented the *Alliance* training, with the aim of achieving a balanced distribution across territories.

Participants' gender, profession and area of practice are summarized in **Table 1**. In phase 1, 31 professionals were interviewed: 18 from the school sector (among 63 eligible professionals) and 13 from the municipal sector. In the school sector, three-quarters (6/8) of school directors were women, while the majority of district pedagogical advisors were men (7/10). In the municipal sector, the vast majority of interviews were conducted with local elected officials (9 out of 13), most of whom were women (6 out of 9). In phase 2, 30 professionals were interviewed (among 93 eligible professionals): 25

Table 1. Characteristics of interview participants by phase and overall.

| Characteristics | | | Number of participants (%) | | |
|---|---|---|---|---|---|
| | | | Phase 1 (n=31) | Phase 2 (n=30) | Total (n=61) |
| Gender | Female | | 18 (58%) | 20 (66%) | 38 (63%) |
| | Male | | 13 (42%) | 10 (33%) | 22 (37%) |
| Profession | School | Representatives of the national education system (health advisors, inspectors) | 0 | 7 (23%) | 7 (11%) |
| | | District pedagogical advisors | 10 (32%) | 9 (30%) | 19 (31%) |
| | | Primary school directors – teachers | 8 (26%) | 9 (30%) | 17 (28%) |
| | Munici-palities | Elected officials | 9 (29%) | 5 (17%) | 13 (21%) |
| | | Employees of local authorities; after-school care staff | 4 (13%) | 0 | 4 (7%) |

from the school sector and 5 from the municipal sector, including 20 women. In the school sector, most of school directors and district pedagogical advisors were women (8/9 and 5/9, respectively). The most senior positions – corresponding to inspectors and health advisors – were mainly held by women (5/7). In the municipal sector, the interviews were conducted with local elected officials (n = 5), most of whom were women (4 out of 5). In both phases, professionals from all four intervention departments were represented, across both the school and municipal sectors.

## Data collection

Semi-structured interviews were selected as the most appropriate method to address specific questions aligned with the study objectives, while letting participants the flexibility to share their experiences with the intervention.

Data were collected during two separate periods. The first period involved participants engaged in the intervention during phase 1. Interviews were conducted in 2022 by two investigators (AG and LV, both female) as part of their master's degree theses in Public health and Health promotion and education, respectively, under the supervision of MG, an expert in qualitative research. Participants had no prior relationship with the interviewers. The second period focused on participants involved in phase 2. Interviews were conducted from April to June 2023 by IS (Master's degree, female, project coordinator). Some participants, primarily representatives of the French National Education system, were already acquainted with the interviewer and familiar with the project's goals and methods.

The semi-structured interviews were conducted by telephone or videoconference, enabling participation from stakeholders across all four departments where the intervention was implemented. All interviews were audio recorded and transcribed verbatim. Field notes were also taken. Participants did not provide feedback on the transcripts or final analysis.

The interview guide (S1 File) was initially developed to identify barriers and levers to implementation. Its construction was informed by the literature on school-based health promotion and by the RE-AIM framework [21] to ensure comprehensive coverage of implementation-related dimensions. The qualitative material collected on the intervention, its implementation, and contextual factors, provided sufficient depth to support a subsequent *a posteriori* secondary analysis to identify the key functions and forms and contextual factors (as conceptualized within the FIC model), although they were not originally designed for this purpose. The guide was adapted for each stakeholder group.

## Data analysis

The FIC model analysis relies on joint reflection among various stakeholders [15,16]. First, a thematic content analysis was carried out using QDA Miner Lite and ATLAS.ti software to identify barriers and facilitators to implementation [22]. Subsequently, these qualitative data were analyzed to characterize key functions, forms, and contextual elements as conceptualized within the FIC model. In line with established methods [16,17,23], the analysis was informed not only by qualitative data but also by internal project documentation (*e.g.,* training materials, steering committee reports, and operational notes) and feedback from health promotion consultants (members of the research team), who were responsible for operational coordination, field implementation in schools and municipalities, and direct contact with recipients. These sources supported the triangulation of interview data by providing complementary insights into the theoretical forms of the intervention and clarifying certain organizational and implementation processes. They also contributed to ensuring consistency with field realities and offered additional perspective on the intervention's evolution over time.

The analysis was carried out collaboratively with members of the research team that developed the FIC method (LF, CD), who provided methodological expertise to identify the key functions and forms, and project leaders (IS, VR), who had in-depth knowledge of the intervention. Six working meetings were held between January and March 2024, during which an iterative process of description and validation was followed until consensus was reached. The analysis was further refined through discussions with the health promotion consultants. This process ensured that the description remained

faithful to field realities and employed appropriate terminology. Consequently, the findings were the result of a description process based on the integration of multidisciplinary academic, professional, and experiential knowledge.

Data from phase 1 interviews, internal documentation, and consultants' feedback were used to generate the "theoretical forms" as conceptualized within the FIC model, whereas data from phase 2 interviews were used to produce "observed forms". The "theoretical forms" of phase 1 are therefore distinct from the "observed forms" of phase 2. Phase 2 data did not lead to the emergence of new key functions; rather, they allowed us to confirm their relevance across extended implementation settings and to document how their forms evolved in practice. Only KF1 was modified between phases, shifting from "Multi-institutional and political support" to "Institutional support from the National Education system," as territorial political support declined over time. Comparing theoretical forms with observed forms made it possible to identify and document concrete evolutions in the *Alliance* intervention across implementation phases.

### Ethical considerations

This study received approval from the French Inserm Ethics Evaluation Committee (CEEI / IRB00003888) under reference numbers 19–600 (July 8, 2019) for phase 1 and 23–981, (March 7, 2023) for phase 2 and was conducted in accordance with the principles of the Declaration of Helsinki. Prior to the interviews, all participants were provided with a written description of the study and were informed of their right to withdraw at any time. Oral informed consent was obtained at the start of each interview for phases 1 and 2, after the audio recording had started; participants verbally confirmed their agreement to participate and to be recorded. Consent was thus documented and witnessed through the recording. This procedure was approved by the IRB. To ensure confidentiality, participants' identities and interview content were fully anonymized.

### Results

Seven key functions (KF) were identified across the four years of the *Alliance* intervention and are described below.

- KF 1: Institutional support from the National Education system;

- KF 2: Interprofessional training on health promotion, aimed at teachers and after-school care staff;

- KF 3: Knowledge of, and respect for, the workings of each institution, and adaptation to the needs and constraints of teachers and after-school care staff for intervention systems;

- KF 4: Methodological support and practical resources;

- KF 5: General and cross-functional project coordination;

- KF 6: Bridging fieldwork and research to allow the intervention system to evolve over time;

- KF 7: Clear, shared internal organization of the research team.

Thirty *observed forms* were identified across the seven key functions (**S2 Table**). Evolutions between *theoretical forms* (phase 1) and *observed forms* (phase 2) are highlighted in bold in **S2 Table**, with illustrative examples discussed in the following paragraphs. **Fig 1** provides a schematic representation of the *key functions* and their *observed forms* from phase 2 [23].

**KF 1:** Institutional support from the National Education system was essential for the successful implementation of the *Alliance* intervention. Support from the rectorate facilitated engagement at multiple levels of the National Education system and enabled the designation of contact persons at school, district, departmental, and rectorate levels, facilitating communication with the project leaders throughout the intervention (theoretical and observed forms 1A). This institutional support also allowed district pedagogical advisors to be assigned responsibility for leading (Cantal) or co-leading (Loire,

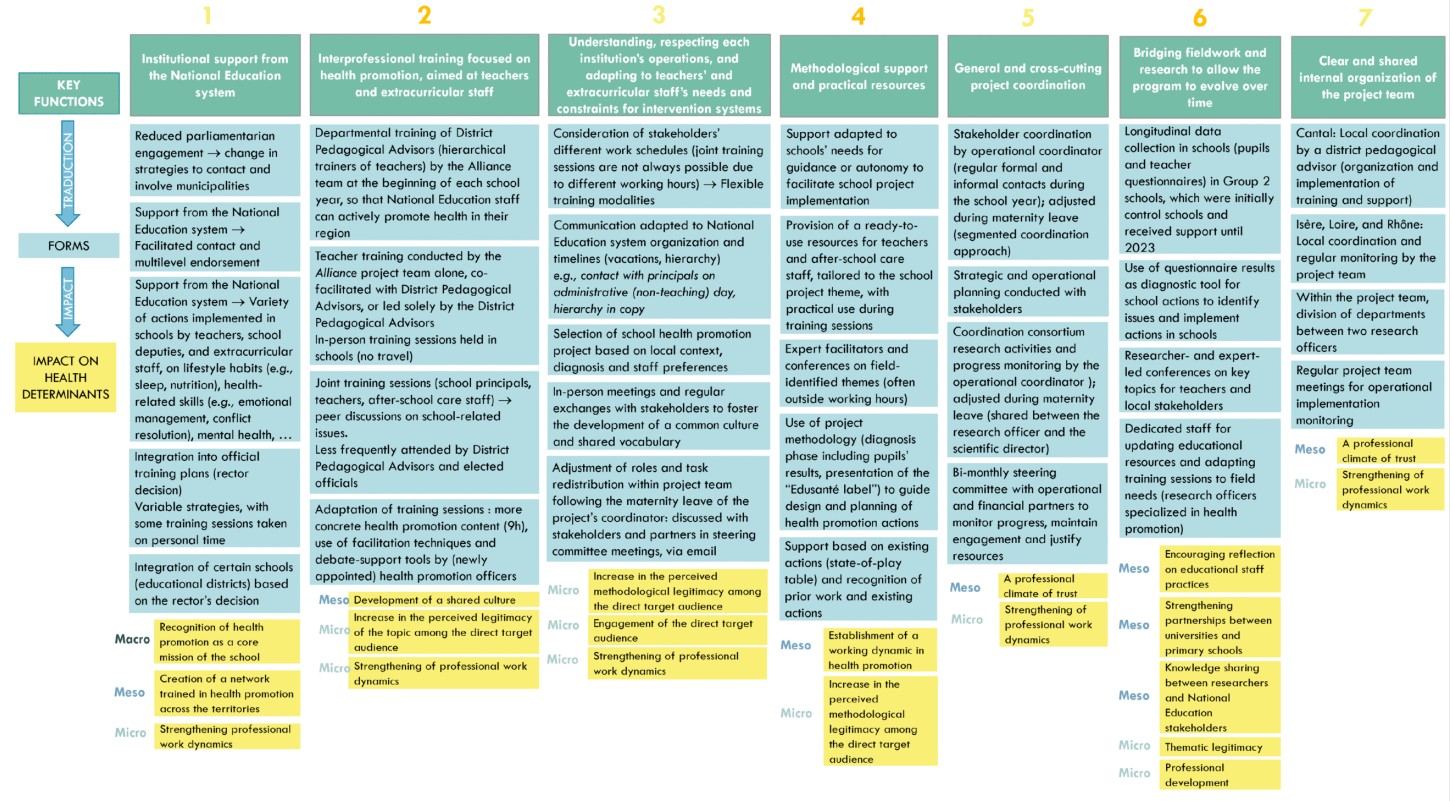

**Fig 1.** "Key functions/implementation/context" diagram illustrating key functions of the *Alliance* intervention and their observed forms (phase 2, 2022-2023).

Isère, Rhône) teacher and after-school care staff training sessions in collaboration with the research team (health promotion consultants). In addition, institutional support made it possible to integrate training into statutory working hours, which was a major facilitator of participation, as it ensured that involvement in Alliance occurred within recognized working time, rather than as an additional voluntary commitment relegated to personal time. Thus, the time dedicated to the *Alliance* intervention was structured and determined from the beginning of the project.

At the municipal level, the involvement of parliamentarians during phase 1 helped secure adherence from municipalities through elected officials (theoretical form 1A). In phase 2, their involvement decreased and coordination with municipalities relied more on the research team and schools, making the support of elected officials more variable (observed form 1A). In several cases, limited municipal support constrained the participation of after-school staff, whose training time was not systematically included in working hours. Therefore, securing the support of elected officials appeared crucial to enable effective collaboration between school and municipal staff – an essential prerequisite for the *Alliance* intervention, which seeks to promote interprofessional work around children's health promotion.

**KF 2:** Teachers and after-school staff received joint training in health promotion whenever possible (2A). These sessions provided a structured space for exchange between professionals who work in the same place but have different missions (2C). Training was delivered either by the research team, by district pedagogical advisors, or jointly, depending on local organization (2B). Over time, the format and content of training evolved to better match field needs and constraints, including reduced training time (from 12 to 9 hours) and greater emphasis on practical, actionable content (observed form 2D). This joint training thus served both as a mechanism for knowledge and skill development and as a means for

fostering collaboration and shared understanding between school and municipal staff, which are key objectives of the *Alliance* intervention.

**KF3:** Knowledge of, and respect for, the functioning of each institution by the project leaders (research team) was essential for adapting the intervention system to the needs and constraints of teachers and after-school care staff. Steering committee meetings and regular exchanges with institutional representatives helped develop a shared culture and supported collective decision-making (theoretical form 3D). During phase 2, exchanges became more frequent, particularly during the signing of the *Alliance* agreement for 2023–2025(observed form 3D). At the municipal level, collaboration was less structured; thus, KF3 was implemented through specific observed forms adapted to municipalities (*e.g.,* informational flyer for municipal stakeholders). This key function was essential to foster the adaptation of intervention mechanisms to institutional constraints and thus the effective acceptability, implementation, and sustainability of the *Alliance* intervention.

**KF 4:** Methodological support and pragmatic resources are essential ingredients of the intervention, providing school and after-school care staff with practical guidance and tools to implement health promotion activities effectively, bridging the gap between theoretical training and field application. Various observed forms were identified. For instance, support was delivered either by the health promotion consultants (Loire, Rhône, Isère) or, by district pedagogical advisors (Cantal department), depending on local organizational realities and institutional resources.

**KF5:** General and cross-cutting coordination was essential to ensure coherence between stakeholders and responsiveness to emerging challenges. Operational coordination played a central role in linking all stakeholders – political representatives, funders, the National Education system, municipalities, field actors, and research partners – through regular formal and informal exchanges. A steering committee convened every two months ensured collective oversight and alignment of the project (theoretical and observed forms 5D).

**KF6:** Bridging research with field practice (health promotion activities in school) was crucial. This KF took different forms. For instance, data collected from pupils were used both for evaluation (research objective) and as a diagnostic tool (practical objective) for teachers (theoretical form 6B) and also for after-school care staff in phase 2 (observed form 6B). Conferences led by researchers or experts on key health-related topics enabled teachers and local stakeholders to access evidence-based knowledge (6C).

**KF7:** A clear internal organization of the research team supported coordination across territories. Local project coordination varied depending on the level of institutional support (7A and 7B): in some departments, district pedagogical advisors led local implementation (organization of training sessions and data collection, provision of training and support to schools), whereas in others, the research team ensured coordination, due to uneven distribution of human resources across territories which limited the time that district pedagogical advisors could dedicate to the *Alliance* intervention in some departments. Of note, the allocation of research team staff responsibilities evolved over time, shifting from a division between schools and municipalities (one professional for all schools, one for all municipalities – theoretical form 7C) to a territorial distribution (one professional responsible for both schools and municipalities within a specific territory – observed form 7C). This last example clearly demonstrates how a key function (KF7), with a specific underlying objective (here, a clear internal organization and defined roles to ensure efficient coordination), can take different forms depending on the context (here, due to staff turnover within the research team).

In addition to key functions and forms, several contextual elements were identified as influencing implementation and categorized across macro, institutional, organizational, school, and individual levels. At the macro level, the post-COVID context both constrained training availability (confinements) and increased awareness of health and well-being issues in schools. Additionally, the 2019 "School of Trust" law, which introduced school self-evaluation processes, created alignment opportunities, as *Alliance*-generated data and initiatives could contribute to these evaluations. At institutional and organizational levels, the inclusion of training time within statutory hours, and political endorsement facilitated implementation, whereas administrative fragmentation, staff turnover, and limited replacement capacity acted as barriers. At the school and

individual levels, facilitators included strong leadership by school directors, prior engagement in health-related projects, interprofessional collaboration, and shared objectives around promoting children's well-being facilitated implementation. Conversely, competing priorities, time constraints among beneficiaries, perceived conceptual ambiguity around health promotion, and tensions with parents were limiting factors.

## Discussion

This study aimed to identify the key functions of the *Alliance* intervention, as conceptualized within the FIC model, to inform and support its transferability. Seven key functions were identified, covering four main dimensions: institutional support (KF1); empowerment of teachers and after-school care staff (inter-professional training (KF2) and provision of methodological support and resources (KF4)); adaptation to beneficiaries' constraints and needs (KF3); and operational alignment and coherent coordination (KF5–7). This description provides an important foundation for the future transfer of the intervention to other settings.

### Key functions of the Alliance intervention in relation to existing school-based health promotion program

The identification of key functions, as conceptualized within the FIC model, is not intended to determine the components of intervention effectiveness, but rather to distinguish transferable theoretical processes that ensure fidelity to the intervention from those aspects of implementation that can be adapted across contexts [14]. However, some key functions – particularly interprofessional training (KF2) and the provision of methodological support and resources (KF4) – are theoretically more related to intervention effectiveness, as they correspond to correspond to key levers identified in the literature as influencing professional practices and the implementation of health promotion activities. The Centers for Disease Control and Prevention's systematic review of school-based health education curricula identifies teacher training, along with enhancing teachers' interest and sense of competence, strengthening alignment with the teaching curriculum, and support for the implementation of evidence-based pedagogical strategies in health education, as key factors associated with effectiveness [24]. Similarly, a meta-analysis of school-based interventions for the prevention of overweight and obesity found that programs were more likely to be effective when teachers had received preparatory training [25]. Although no causal inference can be drawn, the encouraging results observed in phase 1 at both the professional practice and child levels are consistent with the expected role of KF2 and KF4 in supporting changes in professional practices.

KF2 and KF4 are classic key functions in health promotion programs. For example, in the "Lycéen Bouge" program, a similar key function, "Support and training of school advisors" was explicitly identified [26]. In the literature, the empowerment of educational teams is sometimes described as a key function, and in other cases as a form [26,27]. In the present analysis, the FIC model approach provides a clear distinction between what belongs to the key function (essential processes structuring the intervention) and what belongs to the form (context-specific ways in which these processes are operationalized). Distinguishing functions from forms makes it possible to preserve the underlying logic of the intervention while allowing contextual adaptations, which is central when considering transferability across settings.

Key functions related to institutional support, coordination, internal organization create the structural and governance conditions necessary for the intervention to be delivered. Similar functions have been documented in the literature on psychosocial skills development programs in primary and secondary schools. For example, the Affective and Social Development Program (PRODAS) emphasized the importance of "harmonization, sharing of practices, and multiple team discussion opportunities" [27]. This function echoes both the KF3 (knowledge of and respect for institutional functioning) and KF1 (institutional support) of the *Alliance* intervention. Of note, the French educational system is centralized, with a strong role of national and regional authorities. In this context, institutional support at different administrative levels and the involvement of rectorates and inspectors played a major role in facilitating implementation. These elements may not be directly transferable to educational systems with different governance structures or levels of decentralization. International frameworks such as the WHO Health Promoting Schools standards emphasize that the implementation of

school-based health interventions depends on national governance arrangements and institutional contexts and therefore requires adaptation across countries [28].

Furthermore, functions related to coordination and integration (KF5–KF7) appeared particularly important in the context of intervention research. Maintaining a strong link between the research team, institutional partners, and the field actors is essential to 1) ensure that the intervention is adapted to its implementation context, and 2) prevent research activities from being perceived as an additional burden, a key issue in intervention research. Notably, three of the seven key functions relate specifically to coordination and integration with research activities. While structured coordination has been maintained by the research team during the experimental phases of the project, future scale-up may require these responsibilities to be delegated to other stakeholders. Given the central role of coordination in sustaining interventions during scale-up [29], identifying and defining potential new forms for these three key functions in future implementations remains a key challenge.

Deconstructing the intervention into its key functions also illustrates how the *Alliance* intervention operates across different social levels, from micro (individual) to macro levels, to influence health determinants [18]. At the *macro level*, KF1 secures the recognition of health promotion as a core mission of the National Education system. At the *meso level*, KF2 and KF4 support the development of a shared culture and the establishment of collective working dynamics in health promotion, while KF5–7 strengthen partnerships and promote knowledge sharing among stakeholders, and foster a professional climate of trust. At the *micro (individual) level*, KF2 and KF4 enhance individual teachers' and after-school care staff's legitimacy to act in health promotion, as does KF3, for instance by ensuring that health promotion actions are tailored to local needs, contexts, and professionals' preferences.

Finally, the identification of contextual elements influencing implementation provides important guidance for future implementation, as it enables stakeholders to anticipate the institutional and organizational requirements necessary for preserving key functions. For example, securing training time within statutory hours and ensuring adequate teacher replacement capacity emerged as structural prerequisites for maintaining KF2. Similarly, recognizing barriers such as administrative fragmentation or staff turnover allows stakeholders to anticipate challenges.

## Contributions of the FIC model's key functions to study transferability

While school-based health promotion programs are increasing in number, studies in France mainly focus on their effects at the teacher level (*e.g.,* health education practices) and pupil level (*e.g.,* psychosocial skills), including within the present research program and other initiatives such as Explo'Santé [30]. However, they rarely address implementation processes, the conditions required for successful implementation, or the identification of interventions' key functions; and their transferability is not often documented. One of the main contributions of the key functions as conceptualized within the FIC approach is to make explicit dimensions that remain implicit in the daily work of project leaders and are not systematically formalized or highlighted in official documents or communications. Identifying the key functions of the *Alliance* intervention may help future implementers, together with local actors, to 1) be aware of key functions that should be preserved during implementation, 2) anticipate which contextual elements may influence their preservation, and 3) to select appropriate forms adapted to local conditions, according to the specificities of and the knowledge they have about the features of their context, thereby supporting transferability.

The contribution of the FIC model's key functions can be better understood in relation to other frameworks commonly used in implementation and transferability research. Unlike tools that rely on prescriptive checklists to evaluate transferability, such as the ASTAIRE tool [10], the FIC model adopts a different approach, offering a reflexive framework to disentangle the key functions of an intervention (*i.e.*, the theoretical processes that must be preserved to maintain fidelity) from the adaptable forms that can and should be tailored to specific contexts. The model assumes that intervention effectiveness is not determined by key functions alone (in this sense, they differ from what is typically considered the "active ingredients" of an intervention, *i.e.*, the observable, replicable, and irreducible aspects of the intervention [31]) but

emerges from their dynamic interaction with contextual factors during implementation. Moreover, implementation science frameworks [32–34] are widely used to analyze determinants of implementation, the relationships between implementation strategies and outcomes, and conditions for successful delivery. These frameworks help explain why implementation succeeds or fails within a given context and allow anticipation of barriers and facilitators in future implementations. However, they do not explicitly address how an intervention can be transferred across contexts. In this perspective, the FIC analytical framework provides a complementary contribution by focusing on the internal structure of the intervention and clarifying which processes must remain stable across contexts and which may vary depending on local conditions.

## Limitations

This study has several limitations. First, the analysis is partly based on retrospective phase 1 data, which were originally collected to explore barriers and levers to project implementation rather than to identify key functions. This limitation is however mitigated by the inclusion of phase 2 interviews, which were specifically conducted to address this objective. Second, although the sample included a wide range of professional roles, some stakeholder groups were underrepresented. While the FIC model traditionally primarily relies on insights from implementers and researchers [14,15,23], it is important to acknowledge that the perspectives of those directly targeted by the intervention were not fully considered in this study. The absence of pupils (due to the complexity of ethical approval procedures for involving minors) and underrepresentation of certain frontline professionals (teachers other than school directors, municipal after-school care staff who were difficult to contact) limits the understanding of acceptability, feasibility, and perceived relevance of the intervention at the operational level. Third, women were overrepresented among participants. However, this reflects the actual gender distribution in education and child-focused health professions (*e.g.,* 85% of primary school teachers, over 80% of primary school directors and teachers, 71% of pediatricians [35,36]). More broadly, sociological evidence indicates that women are more engaged than men in health, health education (both for themselves and others), and care-related activities, especially those involving children [37,38].

Regarding the FIC approach itself, challenges include the granularity of key functions and the criteria for distinguishing between key functions and forms, which are beyond the appraisal and negotiation of stakeholders. Field actors, who are best positioned to judge which intervention elements can be adapted based on their experience and knowledge of local contexts [39], were actively involved, through interviews and participation of health promotion consultants in the co-construction and revision of the key functions and forms. Nevertheless, it should be noted that some research and project partners were not included in the model's development due to time constraints and competing priorities within the project timeline.

In addition, some researchers (IS, VR) were involved in the design and implementation of the intervention and thus also acted as stakeholders in the project. While this positioning may introduce a potential "judge and party" bias, it also represents a strength by providing in-depth, experience-based insight into the intervention from an insider perspective. This reflects a situated and reflexive approach to knowledge production [40,41]. To mitigate potential bias and circularity in identifying key functions, the analysis was primarily grounded in interview data and supported by iterative discussions with researchers (LF, CD) who were not involved in the intervention's design or implementation. Any disagreements were resolved by consensus. This process led to several refinements: for instance, the initial distinction between municipal- and school-level key functions was removed in favor of a unified approach, where forms indicated whether components pertained to the National Education system or the municipal level; two key functions were merged to improve coherence.

## Perspectives

In this study, the FIC model was applied partially, focusing on the identification of key functions to inform future implementation. The model can be further leveraged to examine social inequalities in health. Analyzing the *Alliance* intervention

through the lens of social inequalities could help anticipate differential effects of the intervention across diverse contexts and support equity-sensitive implementation. Conducting such analyses would require additional, targeted data collection during interviews with both implementers and beneficiaries.

Future research will examine the implementation of the *Alliance* intervention in new groups of schools and municipalities under routine implementation conditions, as opposed to the experimental conditions of phases 1 and 2. These evaluations will generate complementary insights and inform the formalization of practical, context-sensitive recommendations to guide future replication or scale-up, thereby supporting effective and sustainable implementation.

## Conclusion

The characterization of key functions enabled a detailed, structured description of the *Alliance* intervention, integrating perspectives from research teams, field implementers, beneficiaries, and institutional actors. By explicitly identifying key functions and their adaptable forms, this work provides a snapshot of the *Alliance* intervention in its current form, while providing a framework to anticipate and guide its future adaptation across different contexts.

## Supporting information

**S1 File. Interview guide for individual semi-structured interviews with district pedagogical advisors.**
(DOCX)

**S2 Table. Description of theoretical and observed forms of the Alliance intervention.**
(DOCX)

## Acknowledgments

The authors would like to thank Axelle Giron and Lauryne Vacher for data collection and assistance with data analysis; Marine Labruyère and Perrine Ropers for assistance with data analysis; Marine Genton and Amandine Baudot for scientific guidance, as well as Cyrille Isaac Sibylle, Franck Chauvin, Marine Genton and Amandine Baudot *(University Jean Monnet Saint-Etienne, Health, Systemic, Process Research Unit 4129, PRESAGE Insititute)*, Didier Jourdan, Carine Simar, Julie Pironom, Marion Monier *(University Clermont-Auvergne, UR4281 ACTé)*, and Sandy Bernard (*University Claude Bernard Lyon 1, Health, Systemic, Process Research Unit 4129)* for their contributions to the *Alliance* research project. The authors would like to thank all professionals who participated in interviews, as well as, more broadly, schools and municipalities involved in the *Alliance* project.

## Author contributions

**Conceptualization:** Inès Sartori, Cyrille Delpierre, Véronique Régnier Denois.

**Formal analysis:** Inès Sartori, Leslie Fonquerne.

**Funding acquisition:** Cyrille Delpierre, Véronique Régnier Denois.

**Investigation:** Inès Sartori.

**Methodology:** Leslie Fonquerne.

**Validation:** Cyrille Delpierre, Véronique Régnier Denois.

**Writing – original draft:** Inès Sartori, Camille Riera-Navarro.

**Writing – review & editing:** Leslie Fonquerne, Camille Riera-Navarro, Daphné Desmoriaux, Cyrille Delpierre, Véronique Régnier Denois.

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
