## [Decision Letter · Decision Letter 0]

17 Feb 2026

PONE-D-25-54611Key functions for the transferability of a French school-based health promotion intervention: application of the FIC modelPLOS One

Dear Dr. Riera-Navarro,

Thank you for submitting your manuscript to PLOS ONE. After careful consideration, we feel that it has merit but does not fully meet PLOS ONE’s publication criteria as it currently stands. Therefore, we invite you to submit a revised version of the manuscript that addresses the points raised during the review process. The reviewers agree that your study offers a valuable contribution to the literature; however, they have identified several areas where clarification and refinement are needed to strengthen its scientific rigor and overall positioning. They recommend addressing a number of major and minor revisions to improve the manuscript’s clarity, methodological transparency, and alignment with the journal’s standards.

We look forward to receiving your revised manuscript.

Kind regards,

Tahir Turk, PhD

Academic Editor

PLOS One

Journal Requirements:

3. In the ethics statement in the Methods, you have specified that verbal consent was obtained. Please provide additional details regarding how this consent was documented and witnessed, and state whether this was approved by the IRB.

5. In the online submission form, you indicated that the data presented in this study are available from the corresponding author upon reasonable request.

6. We note that the grant information you provided in the ‘Funding Information’ and ‘Financial Disclosure’ sections do not match.

7. Thank you for stating the following financial disclosure:

This work was funded by the French National Cancer Institute (INCa-16818), the French Institute for Public Health Research (IReSP-AAP-2021-273199) as part of the 2021 Research Call on Health-Promoting Services, Interventions, and Policies (supported by CNAM, DGS, Inserm, MILDECA, and Santé publique France); the French Interministerial Mission for Combating Drugs and Addictive Behaviors (MILDECA); and the Auvergne-Rhône-Alpes Regional Health Agency, with funding allocated to Véronique Régnier and Franck Chauvin (University Jean Monnet Saint-Etienne, Health, Systemic, Process Research Unit 4129). The funders had no role in study design, data collection and analysis, decision to publish, or preparation of the manuscript.

The SO-RISP research network, with support from the French National Cancer Institute, received INCa-Cancéropôle GSO support. This research was funded by IReSP as part of the call for structuring actions for research on uses and addictions to psychoactive substances 2021 (IRESP-AAPSPA2021-V1-06).

8. Please amend your authorship list in your manuscript file to include author Véronique Régnier-Denois.

9. Please amend the manuscript submission data (via Edit Submission) to include author Véronique Régnier.

10. We note that there is identifying data in the Supporting Information file <S1_Appendix.docx>. Due to the inclusion of these potentially identifying data, we have removed this file from your file inventory. Prior to sharing human research participant data, authors should consult with an ethics committee to ensure data are shared in accordance with participant consent and all applicable local laws.

-Location data

Additional Editor Comments :

Overall, the study is seen to make a useful contribution by the reviewers, but important clarifications and refinements are needed to strengthen its scientific rigor and positioning.

Reviewers' comments:

Reviewer's Responses to Questions

**Comments to the Author**

1. Is the manuscript technically sound, and do the data support the conclusions?

Reviewer #1: Yes

Reviewer #2: Yes

2. Has the statistical analysis been performed appropriately and rigorously?

Reviewer #1: N/A

Reviewer #2: N/A

3. Have the authors made all data underlying the findings in their manuscript fully available?

Reviewer #1: Yes

Reviewer #2: Yes

4. Is the manuscript presented in an intelligible fashion and written in standard English?

Reviewer #1: Yes

Reviewer #2: Yes

5. Review Comments to the Author

Reviewer #1: The manuscript makes a valuable contribution to the field of PHIR by addressing the challenge of supporting the transferability of interventions, which requires identifying both their invariant components and their contextual adaptations. In this respect, the FIC approach constitutes a robust method for describing complex school-based interventions.

I believe this article could make a meaningful contribution to PLOS One, and I therefore recommend it for publication, subject to minor revisions : a number of comments are offered below, suggesting adjustments that could help clarify the theoretical positioning, methodological choices, and discussion. The questions and suggestions are detailed for each section :

Introduction

- Regarding the IPS indicator, given that the journal has an international readership that may not be familiar with this measure (reference 4 is in French), it might be helpful to further specify what this indicator captures and to indicate the range of values it can take.

- The Alliance intervention was expanded in Phase 2: was there an evaluation of the effectiveness of the intervention in Phase 1 that justified this scale-up?

- In relation to the dialogue with existing conceptual frameworks, you note that “Various models and approaches have been developed to support the assessment of transferability.” These models are only briefly mentioned: it would be useful to clarify how the FIC approach differs from them and/or why it may be more relevant than other methodologies or frameworks for evaluating implementation and fidelity described in the international literature.

- You state that “Transferability involves not only replicating implementation (applicability) but also achieving similar outcomes in new contexts.” It would be helpful to clarify whether the FIC evaluates the replication of implementation and/or outcomes (effectiveness). A similar overlap appears later in the discussion between implementation success and intervention success.

- Finally, why is the potential to assess impacts on equity and social health inequalities—something the FIC method seems well suited to address—not mentioned? This issue is central in PHIR and in discussions of the transferability of complex interventions, yet it does not appear to have been considered in the analysis of the Alliance intervention.

Methods

- The role of Phase 2 data would benefit from being more clearly distinguished (for instance, by specifying whether certain functions emerge only in Phase 2, whether Phase 2 confirms or refines functions retrospectively identified from Phase 1, or whether the only distinction lies between theoretical forms in Phase 1 and observed forms in Phase 2).

- You state that “The guide was developed based on the literature on factors influencing the implementation of school-based health promotion interventions and the RE-AIM framework.” It is somewhat surprising that this theoretical framework was used in the study but is neither mentioned nor described in the theoretical section. Clarifying how RE-AIM contributes to the FIC approach, and how it informed the interview guide in comparison with a “classic” FIC interview grid (if such a generic grid exists), would be helpful. Moreover, if the RE-AIM framework was used beyond the construction of thematic entries in the interview guide, did you also use quantitative indicators to assess implementation and transferability? If so, which ones?

Data Analysis

- You indicate that “model construction was informed not only by qualitative data but also by internal project documentation and feedback from health promotion consultants.” Could you briefly specify what these additional sources contributed? For instance, did they provide new insights, or help triangulate findings from the main qualitative data collection?

- The role of the research team in constructing the key functions would also benefit from greater critical reflexivity, particularly with regard to the risk of circularity (i.e., what “works” being what is already implicitly valued).

Results and Discussion

- The description of key functions and their theoretical and observed forms is clear and well supported by Figure 1, which provides a useful synthesis of the FIC results. However, additional contextual elements (identified barriers and facilitators) might be expected in the results and discussion, as these are clearly shown in the figure but only minimally exploited or discussed in the text.

- The discussion is rich, well-referenced, and well articulated with the literature, enabling meaningful comparisons. It successfully positions the FIC method as a tool for in-depth intervention description and for supporting transferability. However, you focus on certain functions as being critical for intervention effectiveness: it would be useful to clarify whether these key functions contribute to implementation success, intervention effectiveness, or both. If relevant, reporting elements of Alliance’s effectiveness outcomes could help support this argument.

- The discussion could also, where relevant, integrate a comparison of contexts (particularly institutional contexts) in relation to the interventions and the literature mobilized around the key functions of Alliance. In this respect, the specificity of the French educational context could be made more explicit for non-French-speaking readers.

- The perspective on social health inequalities is introduced at the end of the manuscript but would benefit from being more explicitly connected to the study and introduced earlier, including a more timely justification for why this analytical angle was not taken.

- Finally, either in the discussion or the theoretical section, the positioning of the FIC in relation to more prescriptive transferability tools (such as ASTAIRE) is clear and helps to highlight its added value as a reflective framework. However, given the journal’s international readership, it might be useful to briefly discuss how the FIC method could be articulated with other implementation analysis frameworks (e.g., CFIR, PRISM, or the Implementation Research Logic Model), in order to further situate its contribution within the international field.

Thank you for the quality of your work and for this rigorous and insightful contribution.

Reviewer #2: This manuscript addresses a highly relevant and timely issue in population health intervention research: the transferability of complex school-based health promotion interventions. The authors apply the FIC (Functions–Implementation–Context) model to identify key functions underlying the “Alliance for Health” intervention and to distinguish these from adaptable forms across contexts.

The paper is well written, methodologically transparent, and grounded in a solid conceptual and theoretical framework. It provides a detailed and thoughtful description of the intervention and its implementation processes over time. However, the manuscript remains primarily descriptive and conceptual, and several limitations reduce its analytical depth and the strength of its claims regarding transferability.

Overall, the study makes a useful contribution, but important clarifications and refinements are needed to strengthen its scientific rigor and positioning.

Major comments

1. While the manuscript repeatedly refers to transferability and comparable outcomes, the study does not empirically demonstrate that the identified key functions are associated with comparable effects across contexts. The analysis essentially identifies and formalizes key functions a posteriori, based on stakeholders’ accounts and internal project documentation.

As such, the contribution should be framed more clearly as a conceptual and organizational analysis of implementation processes, rather than an empirical assessment of transferability in terms of outcomes.

The authors are encouraged to:

- clarify explicitly that this work does not test causal relationships between key functions and effectiveness;

- moderate statements suggesting that the identified functions ensure or optimize transferability;

rephrase conclusions to emphasize guidance for future implementation rather than evidence of successful transfer.

2. Key functions are co-constructed by researchers, project leaders, and health promotion consultants who were directly involved in the design and implementation of the intervention. While this participatory approach is coherent with the FIC model, it raises a risk of circular reasoning, whereby elements that enabled the intervention to function are retrospectively defined as “key functions”.

The manuscript would benefit from a stronger critical discussion of this limitation, for example:

- Which elements were debated as potentially non-essential?

- Were there functions that varied substantially without jeopardizing implementation?

- How were disagreements resolved during the consensus process?

Clarifying these points would strengthen the analytical robustness of the model construction.

3. The qualitative sample is diverse in institutional terms, but several key perspectives are missing or underrepresented:

- pupils (final beneficiaries),

- teachers other than school directors,

- municipal after-school care staff.

Although the FIC approach traditionally prioritizes implementers and decision-makers, the absence of these voices limits the understanding of acceptability, feasibility, and perceived relevance of the intervention at the operational level. This should be more explicitly acknowledged as a limitation when discussing the generalizability and transferability of the findings.

4. Many key functions (e.g., institutional support, statutory training time, role of rectorates and parliamentarians) are deeply embedded in the French educational and administrative system. This raises questions about: the extent to which these functions are transferable beyond similar national or institutional contexts; which functions are context-specific versus potentially context-agnostic.

The discussion would benefit from a clearer distinction between: functions that are likely to be transferable only within comparable governance systems, and those that may be relevant across different educational or national contexts.

5. Although the study focuses on implementation rather than effectiveness, the repeated reference to “comparable outcomes” creates an expectation that outcomes will at least be discussed in relation to key functions. Currently, outcomes are largely absent from the analysis.

Even a brief, qualitative reflection linking key functions to previously reported results from phase 1 would help align the manuscript’s claims with its empirical content.

6. In the discussion section, the authors could briefly mention how their FIC-based approach complements other French school-based interventions, such as Explo’Santé, which mainly report qualitative outcomes (e.g., changes in psychosocial skills) rather than implementation mechanisms. This would help situate the added value of the present work.

Minor comments

7. The manuscript is relatively long and occasionally repetitive, particularly in the Results and Discussion sections. Some descriptions of key functions could be condensed without loss of clarity.

8. Figure 1 is conceptually informative but difficult to read in its current form (dense content, small font). Improving readability would enhance its usefulness.

9. The distinction between “key functions” and “forms” is central to the paper but could be summarized more succinctly in the Discussion for readers less familiar with the FIC model.

6. PLOS authors have the option to publish the peer review history of their article (what does this mean?). If published, this will include your full peer review and any attached files.

Reviewer #1: **Yes:**Marie Cholley-Gomez

Reviewer #2: No

---

## [Author Response · Author response to Decision Letter 1]

13 Apr 2026

Dear Editor, Dear Reviewers,

Thank you for your valuable feedbacks. Please find below a point-by-point response addressing your queries. Your comments are presented in black and our answers in blue. All changes in the main manuscript have been made using the track changes mode, so that modifications can be easily identified. Line numbers refer to the version of the manuscript without tracked changes.

Best regards,

The authors.

***

Thank you. We have carefully revised the manuscript to comply with PLOS ONE style requirements. First page (title, authors, affiliations):

- The title font formatting has been corrected (italic, bold; first word of the subtitle capitalized).

- The short title has been removed from the manuscript.

- Affiliations have been revised to remove postal codes and reordered from the smallest to the largest institutional entity.

Main body:

- The titles of Figure 1 and Table 1 have been formatted in bold.

- The supplementary files have been renamed in accordance with PLOS ONE style: S1 Appendix is now S1 File, and S2 Appendix is now S2 Table.

Thank you for this reminder. The corresponding author has an ORCID iD: 0009-0008-6654-4947. We experienced technical difficulties when attempting to validate it in Editorial Manager. We will make every effort to resolve this issue in the submission system and remain available to provide any further information if needed.

3. In the ethics statement in the Methods, you have specified that verbal consent was obtained. Please provide additional details regarding how this consent was documented and witnessed, and state whether this was approved by the IRB.

Thank you for this request. Additional details have now been added to the Ethics section of the Methods (lines 238-248). Oral informed consent was obtained at the start of each interview for phases 1 and 2, after the audio recording had started; participants verbally confirmed their agreement to participate and to be recorded. Consent was thus documented and witnessed through the recording. This procedure was approved by the IRB.

Thank you. We carefully checked the revised manuscript and confirm that the ethics statement appears only in the Methods section.

5. In the online submission form, you indicated that the data presented in this study are available from the corresponding author upon reasonable request. All PLOS journals now require all data underlying the findings described in their manuscript to be freely available to other researchers, either 1. In a public repository, 2. Within the manuscript itself, or 3. Uploaded as supplementary information. This policy applies to all data except where public deposition would breach compliance with the protocol approved by your research ethics board. If your data cannot be made publicly available for ethical or legal reasons (e.g., public availability would compromise patient privacy), please explain your reasons on resubmission and your exemption request will be escalated for approval.

Thank you for raising this important point. The data generated in this study consist of qualitative materials, including verbatim interview transcripts in French. Although the transcripts were anonymized, they contain detailed accounts of participants’ professional experiences within a relatively small and identifiable professional environment. As a result, there remains a risk of indirect re-identification through contextual information, even after removal of direct identifiers.

In addition, participants consented to take part in the study, but did not provide consent for unrestricted public sharing of full verbatim transcripts. Public deposition of the complete dataset would therefore not be consistent with the conditions approved by the ethics committee.

For these reasons, the full dataset cannot be deposited in a public repository. We respectfully request an exemption from the public data-sharing requirement for the full dataset on the grounds of participant confidentiality and consent limitation. However, in the interest of transparency and scientific rigor, we remain willing to share de-identified excerpts or relevant materials supporting the findings on an individual basis upon reasonable request.

6. We note that the grant information you provided in the ‘Funding Information’ and ‘Financial Disclosure’ sections do not match. When you resubmit, please ensure that you provide the correct grant numbers for the awards you received for your study in the ‘Funding Information’ section.

Thank you for noting this discrepancy. The funding information has been carefully reviewed and corrected to ensure consistency between the “Funding Information” and “Financial Disclosure” sections in the resubmitted manuscript.

7. Thank you for stating the following financial disclosure:

This work was funded by the French National Cancer Institute (INCa-16818), the French Institute for Public Health Research (IReSP-AAP-2021-273199) as part of the 2021 Research Call on Health-Promoting Services, Interventions, and Policies (supported by CNAM, DGS, Inserm, MILDECA, and Santé publique France); the French Interministerial Mission for Combating Drugs and Addictive Behaviors (MILDECA); and the Auvergne-Rhône-Alpes Regional Health Agency, with funding allocated to Véronique Régnier and Franck Chauvin (University Jean Monnet Saint-Etienne, Health, Systemic, Process Research Unit 4129). The funders had no role in study design, data collection and analysis, decision to publish, or preparation of the manuscript.

The SO-RISP research network, with support from the French National Cancer Institute, received INCa-Cancéropôle GSO support. This research was funded by IReSP as part of the call for structuring actions for research on uses and addictions to psychoactive substances 2021 (IRESP-AAPSPA2021-V1-06).

Please state what role the funders took in the study. If the funders had no role, please state: "The funders had no role in study design, data collection and analysis, decision to publish, or preparation of the manuscript.

Thank you. We have revised the Financial Disclosure statement accordingly and now explicitly state: “The funders had no role in study design, data collection and analysis, decision to publish, or preparation of the manuscript.”

Funding

This work was funded by the French National Cancer Institute (INCa-16818), the French Institute for Public Health Research (IReSP-AAP-2021-273199) as part of the 2021 Research Call on Health-Promoting Services, Interventions, and Policies (supported by CNAM, DGS, Inserm, MILDECA, and Santé publique France); the French Interministerial Mission for Combating Drugs and Addictive Behaviors (MILDECA); and the Auvergne-Rhône-Alpes Regional Health Agency, with funding allocated to Véronique Régnier and Franck Chauvin (University Jean Monnet Saint-Etienne, Health, Systemic, Process Research Unit 4129). The funders had no role in study design, data collection and analysis, decision to publish, or preparation of the manuscript.

The SO-RISP research network, with support from the French National Cancer Institute, received INCa-Cancéropôle GSO support. This research was funded by IReSP as part of the call for structuring actions for research on uses and addictions to psychoactive substances 2021 (IRESP-AAPSPA2021-V1-06). The funders had no role in study design, data collection and analysis, decision to publish, or preparation of the manuscript.

8. Please amend your authorship list in your manuscript file to include author Véronique Régnier-Denois.

Thank you for this careful observation. The authorship list in the manuscript file has been corrected so that the full name Véronique Régnier Denois now appears consistently.

9. Please amend the manuscript submission data (via Edit Submission) to include author Véronique Régnier.

The submission metadata have also been corrected in Editorial Manager so that the author’s full name, Véronique Régnier Denois, appears consistently in the submission system as well as in the manuscript file.

10. We note that there is identifying data in the Supporting Information file <S1_Appendix.docx>. Due to the inclusion of these potentially identifying data, we have removed this file from your file inventory. Prior to sharing human research participant data, authors should consult with an ethics committee to ensure data are shared in accordance with participant consent and all applicable local laws.

-Location data

Thank you for bringing this to our attention. The file in question (now renamed S1 File) contains the interview guide used for data collection. It includes generic questions addressed to one category of participants (district pedagogical advisors) and does not contain individual participant data, names, initials, contact information, dates, identifiers, or other direct personal information. We carefully reviewed the document to ensure that it contains no identifying information, metadata, or hidden content, and we have re-uploaded a clean version accordingly.

Additional Editor Comments:

Overall, the study is seen to make a useful contribution by the reviewers, but important clarifications and refinements are needed to strengthen its scientific rigor and positioning.

Thank you for this overall assessment. We appreciate the reviewers’ and editor’s constructive feedback. We have addressed the requested clarifications and refinements in the revised manuscript (see detailed responses in the sections above).

***

Reviewers’ comments:

Reviewer #1:

The manuscript makes a valuable contribution to the field of PHIR by addressing the challenge of supporting the transferability of interventions, which requires identifying both their invariant components and their contextual adaptations. In this respect, the FIC approach constitutes a robust method for describing complex school-based interventions.

I believe this article could make a meaningful contribution to PLOS One, and I therefore recommend it for publication, subject to minor revisions: a number of comments are offered below, suggesting adjustments that could help clarify the theoretical positioning, methodological choices, and discussion.

The questions and suggestions are detailed for each section:

Introduction

- Regarding the IPS indicator, given that the journal has an international readership that may not be familiar with this measure (reference 4 is in French), it might be helpful to further specify what this indicator captures and to indicate the range of values it can take.

Thank you for this helpful suggestion. We have clarified the nature of the IPS and specified its range of values for readers unfamiliar with this French indicator (lines 58-61).

- The Alliance intervention was expanded in Phase 2: was there an evaluation of the effectiveness of the intervention in Phase 1 that justified this scale-up?

The extension of the Alliance intervention in Phase 2 was informed by encouraging preliminary findings from Phase 1, including favorable evolutions in professionals’ practices and child-level outcomes; however, these results are not yet published. We have clarified this point in the Introduction (lines 66-74)

- In relation to the dialogue with existing conceptual frameworks, you note that “Various models and approaches have been developed to support the assessment of transferability.” These models are only briefly mentioned: it would be useful to clarify how the FIC approach differs from them and/or why it may be more relevant than other methodologies or frameworks for evaluating implementation and fidelity described in the international literature.

Thank you for this important comment. We agree that the manuscript needed to more clearly position the FIC approach in relation to existing frameworks. We have therefore expanded both the Introduction (lines 89-94) and the Discussion (lines 460-78) to clarify how the FIC model differs from reporting tools and implementation frameworks.

- You state that “Transferability involves not only replicating implementation (applicability) but also achieving similar outcomes in new contexts.” It would be helpful to clarify whether the FIC evaluates the replication of implementation and/or outcomes (effectiveness). A similar overlap appears later in the discussion between implementation success and intervention success.

Thank you for this helpful remark. We agree that the initial formulation introduced ambiguity on what the FIC model actually evaluates. To clarify this point, we now explicitly state that the FIC model approach is grounded in the assumption that the transferability of an intervention - including its potential to reproduce effects - relies, among other conditions, on the transfer of its key functions. While this condition alone is not sufficient, it constitutes a necessary foundation for supporting implementation in new contexts. Accordingly, it is not intended to ensure the effectiveness of the intervention itself (i.e., its ability to achieve the targeted outcomes), but rather to support the effectiveness of its transfer (i.e., success of its implementation in a different context). This clarification has been added to the Introduction section (lines 99-105) We have also revised the manuscript in several places to avoid saying that transferability implies achieving similar outcomes in new contexts, as this is not the focus of the study and the FIC model.

- Finally, why is the potential to assess impacts on equity and social health inequalities—something the FIC method seems well suited to address—not mentioned? This issue i

---

## [Editor Report · Decision Letter 1]

15 Apr 2026

Key functions for the transferability of a French school-based health promotion intervention

PONE-D-25-54611R1

Dear Dr. Riera-Navarro,

Thank you for your detailed revisions. The manuscript and accompanying response document now fully address the concerns raised by the reviewers and the journal. We’re pleased to inform you that your manuscript has been judged scientifically suitable for publication and will be formally accepted for publication once it meets all outstanding technical requirements.

Within one week, you’ll receive an e-mail detailing any further amendments. When these have been addressed, you’ll receive a formal acceptance letter and your manuscript will be scheduled for publication.

Kind regards,

Tahir Turk, PhD

Academic Editor

PLOS One

---

## [Editor Report · Acceptance letter]

PONE-D-25-54611R1

PLOS One

Dear Dr. Riera-Navarro,

I'm pleased to inform you that your manuscript has been deemed suitable for publication in PLOS One. Congratulations! Your manuscript is now being handed over to our production team.

Kind regards,

on behalf of

Dr. Tahir Turk

Academic Editor

PLOS One